## Research Article

COVID-19 pandemic; mental health symptoms; pharmacists; psychiatric hospitals

**Corresponding authors:**
Feng Jiang and Huanzhong Liu;
Emails: fengjiang@sjtu.edu.cn;
zhouxqlulu@126.com

Ling Zhang and Zhenkuo Li are co-first authors of the article.

# Mental health symptoms and their associated factors among pharmacists in psychiatric hospitals during the early stage of the COVID-19 pandemic

Ling Zhang[1,2] , Zhenkuo Li[1,3], Mengdie Li[1,2], Yating Yang[1,2], Michael Hsu[4,5], Lei Xia[1,2], Tingfang Liu[6], Yuanli Liu[6], Feng Jiang[7,8], Huanzhong Liu[1,2] and Yi-lang Tang[4,5]

[1]Department of Psychiatry, Chaohu Hospital of Anhui Medical University, Chaohu, China; [2]Department of Psychiatry, School of Mental Health and Psychological Sciences, Anhui Medical University, Hefei, China; [3]Department of Psychiatry, The Fifth People's Hospital of Xiangtan City, Xiangtan, China; [4]Department of Psychiatry and Behavioral Sciences, Emory University, Atlanta, GA, USA; [5]Mental Health Service Line, Atlanta VA Medical Center, Decatur, GA, USA; [6]School of Health Policy and Management, Chinese Academy of Medical Sciences & Peking Union Medical College, Beijing, China; [7]Institute of Healthy Yangtze River Delta, Shanghai Jiao Tong University, Shanghai, China and [8]School of International and Public Affairs, Shanghai Jiao Tong University, Shanghai, China

## Abstract

As frontline workers, pharmacists often face significant work stress, especially in psychiatric settings. A multicenter cross-sectional design was conducted in 41 psychiatric hospitals. The Depression, Anxiety and Stress Scale–21 (DASS-21) was used to measure the mental health of 636 pharmacists. We also collected demographic data and work-related variables. The prevalence of depression, anxiety and stress was 20.60%, 22.96% and 8.96%, respectively. Multivariate logistic regression showed that several common factors were associated with depression, anxiety and stress, including professional identity (odds ratio [OR] = 0.132, 0.381 and 0.352) and verbal violence (OR = 2.068, 2.615 and 2.490). Those who were satisfied with their job were less likely to develop depression (OR = 0.234) or anxiety (OR = 0.328). We found specific factors associated with mental health. Older age (OR = 1.038) and perceived negative impact (OR = 2.398) of COVID-19 on medical work were associated with anxiety, and those with frontline experience with COVID-19 patients (OR = 2.306) were more likely to experience stress. More than one-fifth of pharmacists in psychiatric hospitals experienced symptoms of depression or anxiety during the pandemic, highlighting the need for policy change to improve workplace conditions and psychological well-being for this professional group.

## Impact statement

Pharmacists represent the third largest group of healthcare workers in the world after doctors and nurses. However, pharmacists are often overlooked and may be marginalized when it comes to protocols aiming at protecting healthcare workers and improving their well-being during the early stage of the COVID-19 pandemic. The study is to explore the mental health status and its associated factors among pharmacists working in psychiatric hospitals in the early stage of the COVID-19 pandemic. Based on a national multicenter sample of pharmacists from 41 psychiatric hospitals, we found that more than one-fifth of pharmacists in psychiatric hospitals experienced symptoms of depression or anxiety during the pandemic, which calls for the attention of hospital administrators. Targeted interventions might include improving job satisfaction, promoting a sense of professional identity and reducing workplace verbal violence. Factors associated with COVID-19 also need to be taken into account when developing prevention and intervention measures. Our work was the first study to focus on the mental health status of a neglected population of pharmacists in psychiatric hospitals. Even though the pandemic has been declared to be over, our research provide insight into future studies and initiatives to hospital policymakers for improving the mental health of pharmacists in psychiatric hospitals.

## Background

During the early stage of the pandemic, the world faced an unprecedented health crisis, which not only posed a direct threat to people's physical health but also had a profound impact on their

mental health. The COVID-19 pandemic may lead to increased workload and stress among healthcare workers (Young et al., 2021). Work-related stressors were reported to be associated with negative effects on the mental and emotional health of healthcare workers (Khanal et al., 2020). The prevalence of anxiety, depression, sleep problems and pain during the epidemic also requires attention (Muller et al., 2020). Researchers also call for prioritization of mental health and well-being among health workers (Søvold et al., 2021).

Pharmacists represent the third largest group of healthcare workers in the world after doctors and nurses (Chan & Wuliji, 2009) and are the most accessible healthcare professionals to patients (Jones et al., 2017). However, pharmacists are often overlooked and may be marginalized when it comes to protocols aiming at protecting healthcare workers and improving their well-being during the early stage of the COVID-19 pandemic (Elbeddini et al., 2020). Recently, a national study conducted in Nigeria showed that the COVID-19 pandemic has significantly impacted the mental health and well-being of pharmacists leading to fatigue and burnout. These symptoms are shown to be associated with work-related factors such as work environment and full-time employment (Hedima et al., 2022). A study on community pharmacists showed that 44.8% of respondents reported depression, 53.3% reported anxiety, and 25.4% reported stress during the early stage of the COVID-19 pandemic (Samir AlKudsi et al., 2022). Low job satisfaction, high workload and lack of recognition and respect are the reasons why some pharmacists have poor mental health and even want to leave the profession (Pharmaceutica, 2020). The increased workload and role burden, drug shortage and workplace harassment negatively impacted the psychological health of pharmacists (Elbeddini et al., 2020), which also demonstrated the importance of mental health in the workplace concerning the quality of services and the correct use of medications.

Clinical pharmacy services in China did not start until 2005 and remain in the preliminary stages of development (Chen et al., 2015). Unlike in countries such as the United States and Australia, the role of pharmacy services remains under-recognized and is considered an auxiliary service. Pharmacists in China are often assigned to perfunctory roles such as compounding and dispensing medicines (Sun et al., 2016; Li & Li, 2018). Similar to pharmacists in other countries (Irwin et al., 2013), there is a general lack of understanding and respect for the role of pharmacists in China, and pharmacists are often perceived as merely medication dispensers and undervalued in Chinese medical institutions (Chen et al., 2015).

Patients with mental illness often require long-term follow-up and poorly adhere to medication regimens, leading to added stress and challenges among pharmacists at psychiatric hospitals. 41.4% of pharmacists reported feeling less comfortable with medication counseling for mental health-related medications than cardiac medications (Goodman et al., 2017). Integrating pharmacists into clinical rounds in psychiatric hospitals can reduce drug-related problems and improve the outcomes for patients with mental disorders (Stuhec & Tement, 2021). Pharmacists in psychiatric hospitals play a key role in the interdisciplinary psychiatric clinical team, providing comprehensive medication management, improving the quality of care and promoting patient safety (Lu et al., 2021a). The misunderstanding of psychiatric care and the stigma of the illness may make psychiatric patients more reluctant to seek psychiatric care (Lahariya et al., 2010), and this may affect the morale and attitude of pharmacists working in psychiatric hospitals. A recent study highlighted the importance of including pharmacists in the home care plans of patients with mental

disorders, thereby improving medication safety (Farag et al., 2022). Therefore, supporting the mental health and well-being of pharmacists in psychiatric hospitals is of great importance for the well-being of patients.

While prior studies have examined the mental health of psychiatric nurses and psychiatrists, little has been done on the mental health of pharmacists in psychiatric hospitals. This study aims at exploring the prevalence and related factors of mental health issues among pharmacists in psychiatric hospitals in China during the early stage of the COVID-19 pandemic and may provide insights into future policy interventions.

## Methods

### Study design and participants

From January to March 2021, questionnaires were distributed to healthcare professionals in 41 tertiary psychiatric hospitals through the National Hospital Performance Evaluation Survey. The project was a nationwide study aimed at improving the quality of medical services in China, improving the working conditions of medical workers and providing guidance for formulating national healthcare policies and optimizing resource allocation. The participants of this project include nurses, doctors, pharmacists, psychologists and psychotherapists from 41 psychiatric hospitals. Our research mainly selected data on pharmacists. Surveys were distributed through an instant messaging app, and each person's social media account could only complete the survey once to avoid duplicate submissions.

We invited 880 pharmacists from 41 tertiary psychiatric hospitals in 28 provinces to complete an online survey and ultimately received 815 questionnaires, with a response rate of 92.61%. Finally, 636 questionnaires were included in the final study analysis after removing logical errors. The study was approved by the Ethics Committee of Chaohu Hospital affiliated with Anhui Medical University. We used an anonymous survey in this study. Before completing the questionnaire, respondents were informed of the privacy protection measures, which assured them that their privacy was protected. Participant responses were kept confidential throughout the data gathering and analysis process. Completing the questionnaire was voluntary and informed consent was obtained by default once the questionnaire was submitted.

### Measurement

#### Depression, anxiety and stress scale–21 (DASS-21)

We used the Depression, Anxiety and Stress Scale–21 (DASS-21) (Lovibond & Lovibond, 1995) to measure the mental health of all participants, including depression (items 3, 5, 10, 13, 16, 17 and 21), anxiety (items 2, 4, 7, 9, 15, 19 and 20) and stress (items 1, 6, 8, 11, 12, 14 and 18). The DASS-21 scale includes 20 items with a 4-point scoring system from 0 (*Did not apply to me at all*) to 3 (*Applied to me very much*). The standard scores for depression, anxiety and stress are obtained by multiplying the sum of the respective seven items scored by 2. The higher the score, the higher the negative emotion level of the individual. The cut-off of this scale was as follows: depression (normal: 0–9, mild: 10–13, moderate: 14–20, severe: 21–27 and extremely severe: ≥28); anxiety (normal: 0–7, mild: 8–9, moderate:10–14, severe: 15–19 and extremely severe: ≥20) and stress (normal: 0–14, mild: 15–18, moderate: 19–25, severe: 26–33 and extremely severe: ≥34). This scale has been used

in the general Chinese population (Liu et al., 2021), and Cronbach's alpha of the total scale in this study was 0.922.

### Minnesota satisfaction questionnaire (MSQ)

We used the Minnesota Satisfaction Questionnaire (MSQ) short form to measure job satisfaction (Hirschfeld, 2000; Rannona, 2003). This scale contains a total of 20 items. A Likert scale is used to score 5 levels; 5 means very satisfied and 1 means very dissatisfied. According to the score of each item, job satisfaction can be divided into intrinsic satisfaction (items 1, 2, 3, 4, 7, 8, 9, 10, 11, 15, 16 and 20), extrinsic satisfaction (items 5, 6, 12, 13, 14 and 19) and general satisfaction (items 1, 2, 3, 4, 5, 6, 7, 8, 9, 10, 11, 12, 13, 14, 15, 16, 17, 18, 19 and 20). A general job satisfaction score of ≥80 points was defined as "satisfied" and otherwise as "dissatisfied" (Jiang et al., 2019). The Cronbach's alpha of the total scale in this study was 0.895.

### Author-designed questionnaire assessing demographic and work-related characteristics

Based on the existing literature, our previous research and expert opinions, we developed an author-designed questionnaire. We used the author-designed questionnaire to collect demographic characteristic data of all participants, including age, gender, marital status, educational level and work-related factors, including working hours per day, monthly night shift, annual income, sense of professional identity and workplace violence. The following question assessed the sense of professional identity: "Do you identify with your current occupation (yes/no)?"; Workplace violence was assessed by the following two questions, including verbal violence and physical violence: "What is the frequency of verbal violence (use of abusive, vilified, defiant, mock, *etc.,* stigmatizing discriminatory language but no physical contact) you've experienced in the last year from patients?"; "What is the frequency of physical violence (violence with physical contact or the use of a weapon) from patients you experienced in the last year? (none; <12 times/year; 1 time/month; 2–3 times/month; 1 time/week; 2–5 times/week; almost every day)". The following two questions assess pandemic-related factors: "Did you have frontline experience with COVID-19 patients (yes/no)"; "What do you think the impact of pandemic on your medical work (negative impact: intend to leave and change the profession; neutral impact: no impact to the profession; positive impact: prefer to work as a pharmacist and love this profession more)". Before the survey was available to the large sample, a pilot study was pretested in a small sample of about 300 participants including doctors, nurses, psychologists, psychotherapists and pharmacists. Changes were made based on the feedback to ensure that the questionnaire was easy to understand and complete.

### Data analysis

We used the software SPSS 23.0 (IBM Corporation, Armonk, NY, USA) for data analysis. The Kolmogorov–Smirnov test was used to detect the normality of continuous variables. The data were not normally distributed so all the continuous variables were expressed as quartile spacing (IQR). The continuous and categorical variables were analyzed by the Mann–Whitney U test and the chi-square test, respectively, between the different groups (depression group *vs.* the non-depression group, anxiety group *vs.* non-anxiety group, stress group *vs.* non-stress group). Finally, we used the multivariate

logistic regression analysis to identify significant predictive variables associated with depression, anxiety and stress. In the multivariate logistic regression model, depression, anxiety and stress were the dependent variables, and the independent variables were the variables that showed significant differences between the three different groups in the univariate analysis ($P < 0.2$). All independent variables were analyzed for covariance before entering the multivariate logistic regression model. A two-tailed *P*-value of 0.05 was considered to be statistically significant except for the univariate analysis.

## Results

### Demographics and characteristics of pharmacists in psychiatric hospitals

As seen in Table 1, 636 pharmacists were enrolled in our study, of which 29.40% were male. The average age of the whole sample was 36 (31, 43), and the majority of them were married (79.09%). Most participants (64.31%) held a bachelor's degree and nearly half of the pharmacists (43.55%) had a mid-level professional title. The average annual income of the whole sample was 10 (8, 15) (Ten thousand RMBs).

### Overall mental health (depression, anxiety and stress) and work-related factors of pharmacists in psychiatric hospitals

According to the results of the Depression, Anxiety and Stress Scale–21 (DASS-21), the prevalence of depression, anxiety and stress symptoms was 20.60% (95% confidence interval [CI]: 17.4–23.7%), 22.96% (95% CI: 19.7–26.2%) and 8.96% (95% CI: 6.7–11.2%), respectively in the whole sample. And 10.69% of the whole sample had frontline experience with COVID-19 patients. "Regarding the perceived impact of COVID-19 on medical work, 39.78% of them reported that they preferred to work as a pharmacist and love this profession more (positive impact), and 51.73% reported intending to leave and change the profession (negative impact), only 8.49% of them reported that there was no impact of COVID-19 on their profession (neutral impact)". According to the results of the Minnesota Satisfaction Questionnaire (MSQ), only 27.52% of pharmacists were satisfied with their jobs. The scores of each dimension of the MSQ were as follows: the score of intrinsic satisfaction was 44 (39, 48), the score of extrinsic satisfaction was 21 (18, 24) and the score of general satisfaction was 73 (64, 80). In addition, the majority (94.50%) had a sense of professional identity of being a pharmacist. Regarding workplace violence, 36.32% of the sample reported that they had experienced verbal violence and 6.45% had experienced physical violence in the last year.

### Differences in demographic, work-related factors between three subgroups (depression vs. no-depression group; anxiety vs. no-anxiety group; stress vs. no-stress group)

As shown in Table 1, common significant differences were found in the three subgroups (depression *vs.* no-depression group; anxiety *vs.* no-anxiety group; stress *vs.* no-stress group) in the variables of the impact of COVID-19 on medical work, the score of intrinsic satisfaction, extrinsic satisfaction and general satisfaction, the prevalence of job satisfaction, sense of professional identity and verbal violence (all $P < 0.05$). Regarding workplace violence, a significant difference in physical violence was found in the following two subgroups: anxiety *vs.* no-anxiety group and

**Table 1.** Basic features, occurrences of depressive, anxious and stress symptoms in 636 pharmacists in 41 psychiatric hospitals in China

| Variables | Total sample (n = 636) | Depression (n = 131,20.60%) | No Depression (n = 505) | Z/χ2 | P | Anxiety (n = 146,22.96%) | No anxiety (n = 490) | Z/χ2 | P | Stress (n = 57, 8.96%) | No stress (n = 579) | Z/χ2 | P |
|---|---|---|---|---|---|---|---|---|---|---|---|---|---|
| Age | 36 (31,43) | 37 (33,45) | 35 (31,42) | −1.622 | 0.105 | 37 (32,44.25) | 35 (31,42) | −2.961 | 0.105 | 36 (32,40) | 36 (31,43) | −0.194 | 0.836 |
| Male n (%) | 187 (29.40) | 37 (28.24) | 150 | 0.107 | 0.744 | 40 (27.40) | 147 (30.00) | 0.367 | 0.545 | 19 (33.33) | 168 (29.02) | 0.466 | 0.493 |
| Marital status | | | | | | | | | | | | | |
| Married n (%) | 503 (79.09) | 106 (80.92) | 397(78.61) | 6.804 | 0.033 | 123 (84.25) | 380(77.55) | 3.147 | 0.207 | 48 (84.21) | 455 (78.59) | 1.083 | 0.582 |
| Single n (%) | 113 (17.77) | 17 (12.98) | 96 (19.01) | | | 19 (13.01) | 94 (19.18) | | | 8 (14.04) | 105 (18.13) | | |
| Divorce/widowed n (%) | 20 (3.14) | 8 (6.11) | 12 (2.38) | | | 4 (2.74) | 16 (3.27) | | | 1 (1.75) | 19 (3.28) | | |
| Education | | | | | | | | | | | | | |
| Associate degree or below n (%) | 92 (14.47) | 23 (17.56) | 69(13.66) | 1.747 | 0.417 | 24 (16.44) | 68 (13.88) | 1.010 | 0.604 | 12 (21.05) | 80 (13.82) | 2.279 | 0.320 |
| Bachelor's degree n (%) | 409 (64.31) | 84 (64.12) | 325 (64.36) | | | 89 (60.96) | 320 (65.31) | | | 33 (57.89) | 376 (64.94) | | |
| Master or higher n (%) | 135 (21.23) | 24 (18.32) | 111 (21.98) | | | 33 (22.60) | 102 (20.82) | | | 12 (21.05) | 123(21.24) | | |
| Professional title | | | | | | | | | | | | | |
| Junior n (%) | 250 (39.31) | 46 (35.11) | 204 (40.40) | 1.921 | 0.383 | 24 (16.44) | 68 (13.88) | 1.010 | 0.604 | 12 (21.05) | 80(13.82) | 2.279 | 0.320 |
| Middle n (%) | 277 (43.55) | 64 (48.85) | 213 (42.18) | | | 89 (60.96) | 320 (65.31) | | | 33 (57.90) | 376(64.94) | | |
| Senior n (%) | 109 (17.14) | 21 (16.03) | 88 (17.42) | | | 33 (22.60) | 102 (20.82) | | | 12 (21.05) | 117(20.21) | | |
| Working hours per day | 8 (7.5,8) | 8 (8,8) | 7.5 (8,8) | −0.955 | 0.340 | 8 (7.88,8) | 8 (7.5,8) | −0.639 | 0.523 | 8 (8,8) | 7.5 (8,8) | −1.676 | 0.094 |
| Monthly night shift | 0 (0,3) | 0 (0,3) | 0 (0,2) | −2.066 | 0.039 | 0 (0,3) | 0 (0,2) | −1.541 | 0.123 | 0 (0,3) | 0 (0,3) | −0.725 | 0.468 |
| Annual income (Ten thousand RMBs) | 10 (8,15) | 10 (7,16) | 10 (8,15) | −1.021 | 0.307 | 10 (7,15) | 10 (8,15) | −1.021 | 0.307 | 10 (7,15) | 10 (8,15) | −0.304 | 0.761 |
| Frontline experience with COVID-19 patients (Yes) n (%) | 68 (10.69) | 16 (12.21) | 52(10.30) | 0.400 | 0.527 | 19 (13.01) | 49(10.00) | 1.070 | 0.301 | 10 (17.54) | 58(10.02) | 3.079 | 0.079 |
| Perceived impact of COVID-19 on medical work | | | | | | | | | | | | | |
| Positive (%) | 253 (39.78) | 63 (48.09) | 190 (37.62) | 25.067 | <0.001 | 59 (40.41) | 194 (39.59) | 19.796 | <0.001 | 28 (49.12) | 225(38.86) | 8.655 | 0.013 |
| Neutral (%) | 54 (8.49) | 22 (16.79) | 32 (6.34) | | | 25 (17.12) | 29 (5.92) | | | 9 (15.79) | 45(7.77) | | |
| Negative (%) | 329 (51.73) | 46 (35.11) | 283 (56.04) | | | 62 (42.47) | 267 (54.49) | | | 20 (35.09) | 309(53.37) | | |
| Intrinsic satisfaction | 44 (39, 48) | 37 (34,43) | 45 (40,48) | −9.989 | <0.001 | 40 (35,45) | 45 (40,48) | −7.983 | <0.001 | 38 (32, 43) | 45 (39, 48) | −4.922 | <0.001 |
| Extrinsic satisfaction | 21 (18, 24) | 18 (15,21) | 22 (19,24) | −9.448 | <0.001 | 18 (16, 21.25) | 22 (19,24) | −7.655 | <0.001 | 19 (15, 22) | 21 (18, 24) | −4.563 | <0.001 |
| General satisfaction | 73 (64, 80) | 60 (55,71) | 75 (67,80) | −10.052 | <0.001 | 64.5 (57,73) | 75 (67,80) | −8.269 | <0.001 | 64 (54.5, 72.5) | 73 (65,80) | −5.003 | <0.001 |
| Satisfied with job n (%) | 175 (27.52) | 10 (7.63) | 165 (32.67) | 32.699 | <0.001 | 16 (10.96) | 159 (32.45) | 26.046 | <0.001 | 9 (15.79) | 166(28.67) | 4.137 | 0.038 |
| Sense of professional identity (Yes) | 601 (94.50) | 106 (80.92) | 495 (98.02) | 58.514 | <0.001 | 127 (86.99) | 474 (96.73) | 20.556 | <0.001 | 47 (82.46) | 554(96.68) | 17.455 | <0.001 |
| Workplace violence | | | | | | | | | | | | | |
| Verbal violence (Yes) n (%) | 231 (36.32) | 70 (53.44) | 161(31.88) | 20.893 | <0.001 | 84 (57.53) | 147 (30.00) | 36.871 | <0.001 | 35 (61.40) | 196(33.85) | 17.031 | <0.001 |
| Physical violence (Yes) n (%) | 41 (6.45) | 15 (11.45) | 26((5.15) | 2.424 | 0.120 | 19 (13.01) | 22 (4.49) | 13.551 | <0.001 | 10 (17.54) | 31 (5.35) | 12.785 | <0.001 |

stress *vs.* no-stress group (all *P* < 0.05). In addition, significant differences in marital status and monthly night shift were found only between the depression and no-depression groups (all *P* < 0.05).

### Factors associated with mental health (depression, anxiety and stress)

The collinearity analysis showed that the variance inflation factor of all independent variables was less than 3. Three multivariate logistic regressions were performed separately to explore factors related to depression, anxiety and stress among pharmacists in psychiatric hospitals. As seen in Table 2, we found that several common factors were associated with all three of depression, anxiety and stress, including a sense of professional identity (OR = 0.132, *P* < 0.001, 95%CI: 0.058–0.301; OR = 0.381, *P* = 0.015, 95%CI: 0.175–0.827; OR = 0.352, *P* = 0.024, 95%CI: 0.142–0.872) and verbal violence (OR = 2.068, *P* = 0.001, 95%CI: 1.321–3.240; OR = 2.615, *P* < 0.001, 95%CI: 1.722–3.971; OR = 2.490, *P* = 0.004, 95%CI: 1.345–4.608). In addition, pharmacists in psychiatric hospitals who were satisfied with jobs (OR = 0.234, *P* < 0.001, 95%CI: 0.116–0.472; OR = 0.328, *P* < 0.001, 95%CI: 0.184–0.584) were less likely to develop depression and anxiety.

However, specific factors associated with depression, anxiety and stress also emerged. Pharmacists in psychiatric hospitals with older age (OR = 1.038, *P* = 0.006, 95%CI: 1.011–1.065) were significantly more susceptible to developing depression and those who perceived negative impact (OR = 2.398, *P* = 0.010, 95%CI: 1.234–4.656) (relative to neutral impact) of COVID-19 on medical work were more prone to suffer from anxiety. In addition, pharmacists in psychiatric hospitals with frontline experience with

COVID-19 patients (OR = 2.306. *P* = 0.025, 95%CI: 1.113–4.780) were more likely to develop stress.

### Discussion

To the best of our knowledge, this was the first multicenter sample study of mental health symptoms among pharmacists in psychiatric hospitals in the early stage of the COVID-19 pandemic. Although our study was completed during the early stage of pandemic, studies suggest that mental health outcomes will likely worsen after the COVID-19 pandemic (Fiorillo & Gorwood, 2020). A study showed 17% of community pharmacists reported post-traumatic stress symptoms after the COVID-19 pandemic (Lange et al., 2020). Our findings inform potential preventative measures to improve mental health among pharmacists in psychiatric hospitals.

### Prevalence of depression, anxiety and stress

The results of our study indicated that 20.60% of pharmacists in psychiatric hospitals experienced depression, 22.96% experienced anxiety and 8.96% experienced stress. Rates of mental health symptoms were lower than that of frontline physicians during the early stage of COVID-19 (Elbay et al., 2020) but higher than that of healthcare workers in Italy (Lenzo et al., 2021). Differences in the cut-off values of the scales and the timing of the study surveys may have contributed to the differences in prevalence rates. In addition, the prevalence of stress was low than that in a meta-analysis of frontline healthcare workers studied (Salari et al., 2020); this may be due to the timing of the survey and the fact that only 10.69% of pharmacists were included in this study. Li et al. (2021) also

**Table 2.** Multivariate logistic regression analysis of depressive, anxious and stress symptoms

| Variables | Depression | | | | Anxiety | | | | Stress | | | |
|---|---|---|---|---|---|---|---|---|---|---|---|---|
| | OR | 95% CI | | *P* | OR | 95% CI | | *P* | OR | 95% CI | | *P* |
| | | Lower | Upper | | | Lower | Upper | | | Lower | Upper | |
| Age | 1.038 | 1.011 | 1.065 | 0.006 | 1.021 | 0.998 | 1.046 | 0.076 | – | – | – | – |
| Marital status | | | | | | | | | | | | |
| Single (ref. married) | 0.969 | 0.499 | 1.880 | 0.925 | – | – | – | – | – | – | – | – |
| Divorce/widowed (ref. married) | 2.417 | 0.892 | 6.552 | 0.083 | – | – | – | – | – | – | – | – |
| Monthly night shift | 1.038 | 0.937 | 1.149 | 0.476 | 1.052 | 0.958 | 1.155 | 0.290 | – | – | – | – |
| Working hours per day | – | – | – | – | – | – | – | – | 1.113 | 0.870 | 1.423 | 0.395 |
| Frontline experience with COVID-19 patients (ref. No) | – | – | – | – | – | – | – | – | 2.306 | 1.113 | 4.780 | 0.025 |
| Perceived impact of COVID-19 on medical work | – | – | – | – | – | – | – | – | | | | |
| Positive (ref. neutral) | 0.710 | 0.448 | 1.127 | 0.147 | 1.007 | 0.651 | 1.558 | 0.976 | 0.638 | 0.338 | 1.204 | 0.166 |
| Negative (ref. neutral) | 1.444 | 0.712 | 2.931 | 0.309 | 2.398 | 1.234 | 4.656 | 0.010 | 1.343 | 0.530 | 3.404 | 0.535 |
| Satisfied with job (ref. dissatisfied) | 0.234 | 0.116 | 0.472 | <0.001 | 0.328 | 0.184 | 0.584 | <0.001 | 0.641 | 0.293 | 1.400 | 0.265 |
| Sense of professional identity (Yes) (ref. No) | 0.132 | 0.058 | 0.301 | <0.001 | 0.381 | 0.175 | 0.827 | 0.015 | 0.352 | 0.142 | 0.872 | 0.024 |
| Workplace violence | | | | | | | | | | | | |
| Verbal violence (Yes) (ref. No) | 2.068 | 1.321 | 3.240 | 0.001 | 2.615 | 1.722 | 3.971 | <0.001 | 2.490 | 1.345 | 4.608 | 0.004 |
| Physical violence (Yes) (ref. No) | 1.296 | 0.591 | 2.841 | 0.517 | 1.656 | 0.808 | 3.393 | 0.168 | 1.792 | 0.747 | 4.301 | 0.192 |

*Note*: $R^2$ for depression: 24.2%; $R^2$ for anxiety: 18.5%; $R^2$ for stress: 13.2%.

reported that the stress level of medical staff increased after they worked with COVID-19 patients.

Several factors may be contributing to the relatively high prevalence of mental health issues among surveyed pharmacists. First, pharmacists oftentimes bear the brunt of patients and their families' frustration and grievances as in many instances they are the last stop for patients leaving a healthcare institution or clinic (Wu et al., 2008; Ma et al., 2014). In addition, the centralized procurement or volume-based procurement policies implemented in hospitals may also provide new practice requirements for pharmacists (Lu et al., 2021b). A nationwide shortage of medications has engendered some frustration among patients and families which may partially be directed toward pharmacists. Additional aggression may be directed at pharmacists working in psychiatric hospitals by patients whose psychiatric or substance use conditions have worsened as a result of medication shortages or prolonged waiting times for appointments. In addition, pharmacists may have additional workloads during pandemic such as nucleic acid collection and testing which take time and energy. Many pharmacists were also frontline workers helping patients with COVID-19 (Bukhari et al., 2020). Our study showed that approximately 10.69% of pharmacists had frontline experience with COVID-19 patients. The COVID-19 pandemic led to increased psychosocial stressors among all healthcare workers, including pharmacists (Wang et al., 2021).

### Common contributors to depression, anxiety and stress of pharmacists in psychiatric hospitals

The first factor associated with rates of mental health symptoms (depression, anxiety and stress) among pharmacists in psychiatric hospitals was a sense of professional identity. Pharmacists in psychiatric hospitals with a sense of professional identity may more identify and accept their professional work in psychiatry, which could promote greater motivation to work. In addition to the pathophysiology and treatment of mental illness, social aspects of mental health and stigma should be included in pharmacy education (Morral & Morral, 2017). Pharmacists may be undervalued and underappreciated by the public due to social biases. Many view pharmacists as merely pill dispensers and prescription fillers, which in turn leads to pharmacists experiencing a low sense of professional identity and pride, leading to poor mental health long term. A study conducted by researchers in Bangladesh focusing on mental health among healthcare workers during the early stage of the COVID-19 pandemic reported that having regrets about one's profession was associated with depression and anxiety symptoms (Tasnim et al., 2021). Pharmacists may start doubting their profession during stressful times such as when there are medication shortages or when they are unable to help patients in a timely fashion. The sense of professional identity of healthcare workers could lead to greater psychological distress in the face of major health events (Li et al., 2022). Those pharmacists with a high sense of professional identity may be less prone to burnout during the pandemic and feel a more pronounced sense of mission and responsibility as medical workers. The high sense of professional role identity among pharmacists may be because the COVID-19 pandemic may have acted as a catalyst to strengthen the professional role identity (Watson et al., 2021). In order to enhance the professional identity of pharmacists, efforts can be made to strengthen the recognition of their professional qualifications, strengthen their professional publicity and social recognition and establish a good pharmacist professional organization and

communication platform (Mantzourani et al., 2022; Kearney et al., 2023).

Job satisfaction was associated with a lower prevalence of depression and anxiety among pharmacists in psychiatric hospitals. Research suggests that high job satisfaction positively impacts employee relationships and promotes mental health (Ayele et al., 2020). Higher job satisfaction was reported to be associated with a willingness to stay on the job (Lin et al., 2021). Individuals who positively evaluate work were more likely to be happier and emotionally and psychologically healthier. Furthermore, higher satisfaction with workplace relationships increases social integration, which underscores the importance of maintaining positive work perceptions (Satuf et al., 2018). Pharmacists who were satisfied with their jobs in psychiatric hospitals performed better physically and mentally. Work not only plays a financial support role but also reflects a sense of personal value for many people (Morin et al., 2007). The low job satisfaction of pharmacists is a result of various challenges and stressors, such as the increased pressure from the pandemic and the decline in their income (Alameddine et al., 2022). To enhance job satisfaction, policymakers and hospital administrators should make changes, such as decreasing workload, providing more incentives and fostering a more supportive working environment (Berassa et al., 2021). Moreover, it is essential to monitor the execution of those policy changes after they have been formulated and communicated (Lahariya, 2018).

When it comes to workplace violence, verbal violence was associated with a higher prevalence of mental health symptoms (depression, anxiety and stress) among pharmacists in psychiatric hospitals. Our data revealed a high prevalence of verbal violence among pharmacists in psychiatric hospitals (36.31%). Pharmacists may be experiencing more abuse and harassment from patients since the onset of the COVID-19 pandemic. Interestingly, our data suggest that physical violence was not associated with mental health symptoms among pharmacists. Most healthcare workers who experienced physical violence receive institutional or peer support. In contrast, more than half of healthcare workers who experienced verbal violence claimed that they did not receive organizational protection or peer support, which may lead to more severe post-traumatic symptoms (Nam et al., 2021). We suspect that this may be related to challenges in obtaining evidence of verbal violence, while overt physical violence is usually punished, and relevant legal institutions usually award compensation or even bear criminal punishment. In addition, physical violence may be one-off, and verbal violence may be repeated or even continuous, resulting in more negative psychological outcomes for healthcare workers (Hanson et al., 2015; Arnetz et al., 2017). In psychiatric hospitals, patients with cognitive abnormalities or severe mental disorders may be more likely to enact verbal violence (Yoo et al., 2018). Considering the negative consequences of workplace violence among pharmacists, hospital policymakers should aim to create an environment where psychiatrists can safely and easily report violence and take concrete steps to mitigate recurrence. Lack of understanding of the role of pharmacists in medication safety and prolonged waiting times are common reasons for pharmacists being subjected to workplace violence(Irwin et al., 2013). To address this issue, policymakers should improve hospital delivery systems, shorten patient wait times and promote positive media coverage of pharmacists' contributions. Furthermore, research suggests that hospitals can protect their health workers by providing violence prevention workshops, identifying patients with violent tendencies and enhancing security systems and emergency buttons (Kumari et al., 2022).

### Specific factors associated with anxiety, depression or stress

Our study also identified some factors associated with specific psychiatric symptoms (anxiety, depression or stress) among pharmacists working in psychiatric hospitals. In contrast to a prior study suggesting that age less than 40 is a risk factor for depression (Xiong et al., 2020), our results showed that older age was associated with a higher prevalence of depression among pharmacists in psychiatric hospitals. For older pharmacists, their limited energy may make them unable to cope with complicated work, especially during the pandemic. In addition to the pharmacists' routine work, some pharmacists may also experience added stress due to additional COVID-19-related responsibilities such as supporting nucleic acid testing. The COVID-19 pandemic poses unique challenges for older pharmacists, who may face higher workloads and expectations due to their seniority and expertise. They may also have more anxiety and fear about their susceptibility and exposure to infectious diseases, as they may have weaker immune systems and higher health risks. These factors can contribute to increased stress and depression among older pharmacists.

In terms of work factors related to the pandemic, pharmacists who reported a positive impact of the COVID-19 pandemic on their job perception were less likely to develop anxiety compared to those who reported a neutral impact. Perceiving the positive impact of COVID-19 may be rooted in pharmacists' sense of accomplishment in helping patients and value as a healthcare worker. The present findings accentuate the importance of highlighting the positive impact that pharmacists had as frontline workers during the early stage of the COVID-19 pandemic. Pharmacists with frontline experience with COVID-19 patients were more likely to suffer from stress. Existing research reveals that work-related pressures during the early stage of the COVID-19 pandemic may adversely affect the future career development of pharmacists (Wu et al., 2021). The findings also remind us to provide adequate prejob training for those who need to work on the pandemic's frontline to reduce the contagion risk and concern. In addition, a primary healthcare system may be able to respond effectively to epidemic outbreaks (Lahariya et al., 2020), thereby reducing the impact of pandemic on health workers. Similar to the research conducted in India, we also call for the implementation of more public health-related documents to ensure the mental health of health workers. (Lahariya et al., 2020).

### Limitations and strengths

There were some limitations in this study. First, this study was based on a cross-sectional design, and thus causal conclusions could not be drawn. Second, the questionnaires we used were all self-report questionnaires, and the recall and social desirability biases of the questionnaires could not be avoided. More comprehensive and objective assessment methods are needed to assess the mental health of pharmacists in psychiatric hospitals. Third, our questionnaire distribution was based on a WeChat platform with the principle of voluntary filling. Although the response rate in this study was high, about 7% of pharmacists did not complete the questionnaire, possibly because of mental health symptoms. Therefore, the actual prevalence of mental health symptoms among pharmacists may be higher than the estimated rate in this study. We could not identify the reasons for the non-participation of a small number of pharmacists, as we did not collect demographic data from those who declined to take part in the study. Finally, pharmacists' prior history of mental health disorders was not investigated, and mental health status during the pandemic may also be influenced by existing mental health.

Despite these limitations, there were some strengths in this study. The tertiary psychiatric hospitals involved in this study came from almost all provinces and regions in the Chinese mainland (except Gansu and Xizang). Therefore, the research results are likely to be promoted nationwide. In addition, online anonymous surveys have generated a high response rate and representative information.

## Conclusions

Based on a national multicenter sample of pharmacists from 41 psychiatric hospitals, we found that pharmacists in psychiatric hospitals experience poor mental health, which calls for the attention of hospital administrators and policymakers. Targeted interventions might include improving job satisfaction, promoting a sense of professional identity and reducing workplace verbal violence. In addition, the experience of working with patients with COVID-19 and the perceived impact of COVID-19 should also be taken into account in the process of prevention and intervention.

**Open peer review.** To view the open peer review materials for this article, please visit http://doi.org/10.1017/gmh.2023.71.

**Data availability statement.** The results of all data we used are presented in the manuscript, and data for the current study are available from the corresponding author by reasonable request.

**Acknowledgments.** We would like to thank all the participants of this survey.

**Author contribution.** Substantial contributions to the conception or design of the work: Huanzhong Liu, Feng Jiang. Acquisition, analysis, and interpretation of data for the work: Ling Zhang, ZhenKuo Li, Mengdie Li, Yating Yang, Lei Xia, Tingfang Liu, and Yuanli Liu. Drafting the work: Ling Zhang. Critical revision of the manuscript: Feng Jiang, Michael Hsu, Huanzhong Liu, and Yi-lang Tang. Approval of the final version for publication: All the authors.

**Financial support.** This work was supported by the National Clinical Key Specialty Project Foundation (CN) (no grant number) and Beijing Medical and Health Foundation (grant number MH180924).

**Competing interest.** The authors declare none.

**Ethical standard.** The research protocol was approved by the ethics committee of Chaohu Hospital of Anhui Medical University, the approval number was 202,002-kyxm-02. This work was carried out in accordance with relevant local, national and international guidelines and regulations.

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
