## [Reviewer Report]

Dear Editors:

We would like to submit the enclosed manuscript entitled “Mental health symptoms and their associated factors among pharmacists in psychiatric hospitals during the early stage of the COVID-19 pandemic ”, which we wish to be considered for publication in “Global Mental Health”. No conflict of interest exits in the submission of this manuscript, and manuscript is approved by all authors for publication. I would like to declare on behalf of my co-authors that the work described was original research that has not been published previously, and not under consideration for publication elsewhere, in whole or in part. All the authors listed have approved the manuscript that is enclosed. 

The study was to explore the mental health status and associated factors of pharmacists working in psychiatric hospitals in the early stage of the COVID-19 pandemic. Our study enrolled 636 pharmacists in 41 psychiatric hospitals in China.. We found that more than one-fifth of pharmacists in psychiatric hospitals experienced symptoms of depression or anxiety during the pandemic, and targeted interventions might include improving job satisfaction, promoting a sense of professional identity, and reducing workplace verbal violence.

We deeply appreciate your consideration of our manuscript, and we look forward to receiving comments from the reviewers. If you have any queries, please don’t hesitate to contact me at the address below.

Thank you and best regards.

Yours sincerely,

Huanzhong Liu

E-mail: huanzhongliu@ahmu.edu.cn

---

## [Reviewer Report]

1. Under abstract, methodology to include study period, study site and study type as well. Detailed description is not needed.

2. Introduction includes some concluding points which can be avoided in the first paragraph of the section.

3. Provide the reference or citations at the end of the sentence at all the places in the manuscript.

4. The statement of pharmacists being solely the dispenser of medicines is solely a problem in China? What other countries be facing the same view, can be mentioned as well.

5. The statements should not start with numerical, rather modified that it starts with words.

6. Since the data collection was done using a instant messaging app, describe in detail how the privacy of the study participants was used. Also, need not mention the brand of the app used.

7. Mention about the logical errors that were encountered that led to non-inclusion of more than 20% of the studies. Can this be counted as a non-response rate.

8. Provide the make and manufacturer along with the version of the app/software used in the description.

9. Under results mention the range as found in the study as well at 95% confidence interval in the results section.

- There are a few highly cited papers on health seeking behaviour of people with psychiatric illness which should be refered to and discussed. Such as : Lahariya C, Singhal S, Gupta S, Mishra A). “Pathway of care among psychiatric patients attending a mental health institution in central India.” Indian J Psychiatry 2010; 52: 333-8.

- Lahariya C. Strengthen mental health services for universal health coverage in India. J Postgrad Med. 2018; 64:7-9. doi: 10.4103/jpgm.JPGM_185_17.

-

10.

11. Mention the scales used for diagnoses, impact of COVID-19 on work and job satisfaction in the results section as well.

There is need to discuss recent primary healthcare reforms in low and middle income countries such as India. Author may consider to read and cite following papers:

Lahariya C. ‘Ayushman Bharat’ Program and Universal Health Coverage in India. Indian Pediatr. 2018; 55: 495-506.

Lahariya C. Health & Wellness Centers to Strengthen Primary Health Care in India: Concept, Progress and Ways Forward. Indian J Pediatr. 2020; 87: 916-29.

12. Older age can be a confounder as well when it comes to depression and stress, how was this overcome or balanced out.

13. Under discussion, there is a mention of comparing the study with that from Italy, provide the reason for comparing as the demography and behavioral difference can exist. Rest of the discussion has been well written.

---

## [Reviewer Report]

Thank you for the opportunity to review the manuscript entitled ‘ Mental health symptoms and their associated factors among pharmacists in psychiatric hospitals during the early stage of the COVID-19 pandemic’ which I enjoyed reading.

I compliment the study investigators on their large cross-sectional study with responses from over 600 pharmacists.

Some suggested amends:

Main comments:

Method- any details of the response rate for the survey, and associated limitations in the discussion?

Results/Discussion, - the stress levels prevalence of 8% is quite low, given that depression and anxiety levels are similar to other healthcare workers during covid 19 (e.g. Salari, N., Khazaie, H., Hosseinian-Far, A. et al. The prevalence of stress, anxiety and depression within front-line healthcare workers caring for COVID-19 patients: a systematic review and meta-regression. Hum Resour Health 18, 100 (2020). https://doi.org/10.1186/s12960-020-00544-1).

Maybe this is due to cut offs of scales, but should be discussed further and clarified for the readers.

Discussion: there is significant focus given to a discussion about a low sense of professional identity, which is valid, but this sample has very high (95% with a sense of professional identity), is that really a significant area to be addressed? If the aim is to inform policy change, is your first concern a 5% of respondents issue? I think the authors should contextualize their specific findings more here – and perhaps demote the importance of this specific finding,

In contrast, a striking finding is the low job satisfaction score, less than a third (28%) of respondents are satisfied, and surely, this is your main area to target for mental health and wellbeing? There is a small discussion here but not much detail or local contextualisation. You successfully identify significant levels of depression and anxiety that are found to be associated with low job satisfaction… so how specifically can you address this? Is it role expansion from ‘mere dispensers’, more pay? Greater recognition? What empirically backed or theoretical supported interventions may help?

Again the levels of verbal and physical abuse received by pharmacists in your study are striking, and should be discussed more. Physical abuse regardless of association with depression or anxiety cut offs should be regarded as unacceptable, never mind 6% of respondents, and pharmacists must feel safe at their workplace to accommodate wellbeing – and this should be an action point for policy makers regardless of statistically significant associations? And to a lesser degree verbal abuse, which may proceed physical violence- I would explore this more as it obviously inversely correlates with value and respect of pharmacists.

I think your suggestion that older pharmacists are more likely to experience burnout is contradictory to burnout research risk factors in pharmacists, please consider, or support with systematic reviews.

Finally, I have uncertainty around the stated aim of the manuscript:

Main text “This study aimed to explore the prevalence and related factors of mental health issues among pharmacists in psychiatric hospitals in China during the early stage of COVID-19 pandemic and to consider policy interventions that might promote the mental health of psychiatric pharmacists to potentially improve patient outcomes.”

Abstract “This study aimed to explore the mental health of pharmacists working in psychiatric hospitals during the early stage of the COVID-19 pandemic”

The study certainly meets the aim described in the abstract, but there are no questions in the survey or results about policy interventions, or section in the discussion focused on policy interventions. The study measures prevalence or depression, anxiety and stress and identifies and discussed risk factors, but does not really ‘consider policy interventions’. Either go into depth on policy interventions or narrow the aim of the research.

Minor comment

Abstract:

“The mental health, including depression, anxiety, and stress were collected.” – revise for grammar?

Discussion L31 – change manuscript to ‘study’

Discussion L33 ’ a research = ‘ a study’

Discussion L31 ‘ latter’ stages of pandemic, title = early stages. Reality is about one year into pandemic for data collection, try be consistent about the description of timing -

Table 1

Body violence should be ‘physical violence’

Third column at end missing some data on percentages

---

## [Reviewer Report]

This publication shares the results of a very important research effort. The authors highlight pharmacists as an essential member of the interprofessional care team and demonstrate the importance of monitoring for risks to their mental health. The only suggestion for improvement is to include details on the survey response rate for their study and if unknown, to explain why it was not feasible.

---

## [Reviewer Report]

Dear Authors, 

Based on clear sets of recommendations that I find useful from the two reviewers, might I request to please incorporate some of those insights/ suggestions that you find appropriate in your manuscript ? Could you also address some of the other queries and concerns ? Especially around findings suggesting similar stress/ depression rates amongst other frontline workers and discussion points that you might want to emphasise being located within identity and related frameworks ( job satisfaction, pay etc.) as indicated by Reviewer-2. 

This is an important piece of work and once modified should be published. 

vandana

---

## [Reviewer Report]

Thank you for incorporating the suggested revisions. The additional text p11 L14-22 needs an English language review.

---

## [Reviewer Report]

1. Author to state the relevance of the study in the current times when the pandemic has been declared to be over, Can be mentioned in the impact statement.

2. In abstract, elaborate a little more on the methodology section, regarding the type of study.

3. Background has been deviated towards the mental health and stigma and barriers among general population. It can be modified to stress more about the pharmacists instead. Mention the early stage of pandemic in terms of timeline as well.

4. Mentioned the type of study of the major study from which the data required for the current study was employed.

5. The author designed questionnaire can be mentioned before describing standard questionnaire. Detailed description of questions is not required but can be given as annexure or appendix. Mention if the author designed questionnaire was pretested in a pilot study.

6. Mention the tests of the normality used in the analysis of the collected data.

7. Describe the socio-demographic characteristics of the study population in the text part of the results as well.

8. A section of discussion on common contributors to depression, anxiety and stress, few recommendations have been provided, which can be provided separately. Modify discussion with respect to the results obtained.

9. Include strengths of the study as well.

10. Include a table on basic socio-demographic factors in terms of age group, education, and other factors that are relevant.

11. The pharmacist, specially in primary care need to be mainstreamed. The PHC reforms as part of UHC and some of the work for community engagement in this area need to be cited and discussed. Such as

a. Health & Wellness Centers to Strengthen Primary Health Care in India: Concept, Progress and Ways Forward. Indian J Pediatr. 2020; 87: 916-29. doi: 10.1007/s12098-020-03359-z. Epub 2020 Jul 8. PMID: 32638338

b. Lahariya C, Roy B, Shukla A, Chatterjee M, De Graeve H, Jhalani M, Bekedam H. Community action for health in India: evolution, lessons learnt and ways forward to achieve universal health coverage. WHO South East Asia J Public Health. 2020; 9: 82-91. doi: 10.4103/2224-3151.283002.

c.

d. ’Ayushman Bharat' Program and Universal Health Coverage in India. Indian Pediatr. 2018; 55: 495-506. PMID: 29978817

e. “Pathway of care among psychiatric patients attending a mental health institution in central India.” (Lahariya C, Singhal S, Gupta S, Mishra A). Indian J Psychiatry 2010; 52: 333-8.